# Phytochemical Evaluation and Anti-Inflammatory Potential of *Miconia albicans* (Sw.) Triana Extracts

**DOI:** 10.3390/molecules27185954

**Published:** 2022-09-13

**Authors:** Mariana Inocencio Manzano, Ariana Centa, Alan de Almeida Veiga, Nayara Souza da Costa, Sandro J. R. Bonatto, Lauro M. de Souza, Fhernanda Ribeiro Smiderle

**Affiliations:** 1Instituto de Pesquisa Pelé Pequeno Príncipe, Av. Silva Jardim, 1632, Água Verde, Curitiba CEP 80250-060, PR, Brazil; 2Faculdades Pequeno Príncipe, Av. Iguaçu, 333, Rebouças, Curitiba CEP 80230-020, PR, Brazil; 3Laboratório de Pesquisa Translacional em Saúde, Universidade Alto Vale do Rio do Peixe, Rua Victor Baptista Adami, 800, Centro, Caçador CEP 89500-000, SC, Brazil

**Keywords:** *Miconia albicans*, flavonoids, osteoarthritis, antioxidant, anti-inflammatory activity

## Abstract

The plant *Miconia albicans* (Sw.) Triana has been popularly used in Brazil to treat chronic inflammatory disturbances, such as osteoarthritis. This disease affects 250 million people worldwide, and is associated with intense pain and loss of articular function. There is a lack of information about the phytochemistry and bioactivity of *M. albicans*. Therefore, this study determined the chemical composition of some extracts and evaluated their cytotoxicity, along with their antioxidant and anti-inflammatory, activities using in vitro models. Aqueous and ethanolic extracts were prepared. Afterwards, a liquid–liquid partition was developed using chloroform, ethyl acetate, and *n*-butanol. The extracts were characterized by LC–MS, and their biological activities were evaluated on epithelial cells (Vero), tumoral hepatic cells (Hep-G2), and THP-1 macrophages. LC–MS analyses identified several flavonoids in all fractions, such as quercetin, myricetin, and their glycosides. The crude extracts and *n*-butanol fractions did not present cytotoxicity to the cells. The non-toxic fractions presented significant antioxidant activity when evaluated in terms of DPPH scavenging activity, lipid peroxidation, and ROS inhibition. THP-1 macrophages treated with the *n*-butanol fraction (250 µg/mL) released fewer pro-inflammatory cytokines, even in the presence of LPS. In the future, it will be necessary to identify the phytochemicals that are responsible for anti-inflammatory effects for the discovery of new drugs. In vivo studies on *M. albicans* extracts are still required to confirm their possible mechanisms of action.

## 1. Introduction

Osteoarthritis (OA) affects 250 million people worldwide. This disease is characterized by the progressive destruction of cartilage, caused by inflammation of the synovial membrane, leading to intense pain and loss of articular function [1]. Although it may affect all joints of the body, the most common sites of OA are the hips, knees, hands, and spine. The prevalence of knee OA has doubled since the middle of the 20th century, becoming a major cause of chronic pain and incapacity of the lower limbs [2,3]. This increase is related to the increase in life expectancy, being predicted to be the most prevalent single cause of incapacity until 2030 [4].

The treatment of OA consists of the relief of pain and stiffness by usage of anti-inflammatory drugs and intra-articular glucocorticoids [5,6]. A new diterpenoid molecule derived from the plants of *Sigesbeckia* species has been developed as an anti-inflammatory drug; however, for the treatment of mild-to-severe pain, the most used drugs are non-steroidal anti-inflammatory drugs (NSAIDs) [5,7]. The usage of these drugs is limited to symptomatic episodes, since they do not show disease-modifying effects. Moreover, together with drug interactions, their long-term usage can reduce their effectiveness, and may increase the risk of adverse reactions such as contact dermatitis, itching, or gastrointestinal, kidney, and cardiovascular problems [5,6,8]. Although their efficacy with short- and middle-term usage has been supported by some researchers, more than half of the patients did not present pain relief to comfortable levels after taking such drugs [5,6].

The use of medicinal plants in Brazil is a cultural practice, especially due to the great biodiversity available to the population [9]. The popular knowledge about these medicines is valuable, and contributes to the development of and studies based on ethnopharmacology [9,10]. Among the plants from the Melastomataceae family, *Miconia albicans* (Sw.) Triana, popularly known as “canela-de-velho” in Brazil, has been used for a long time to treat chronic inflammatory disturbances, such as rheumatoid arthritis and other joint issues [11]. Despite its vast popular usage, few studies have been conducted to evaluate this plant’s phytochemical composition and its pharmacological applications [8].

Previous studies have shown biological activities of the extracts of *M. albicans*, such as antioxidant, anti-inflammatory, analgesic, antifungal, antitumor, and antimalarial effects [8,12]. The anti-arthritic activity of *M. albicans* ethanol extract promoted significant pain reduction in murine models of rheumatoid arthritis, without any apparent liver damage. This result was correlated with a reduction in the abundance of the main pro-inflammatory cytokines and chemokines associated with arthritis (i.e., TNF-α, IL-1β, IL-6, and IL-8) [8,12]. However, there is a lack of information comparing their biological effects to the chemical evaluation of the extracts.

Furthermore, the Brazilian Health Regulatory Agency (ANVISA) has banned the marketing of the plant *M. Albicans* as a commercialized phytomedicine due to lack of formal studies on its safety and efficacy, or registration of its uses. Thus, the main objectives of this study were to prepare different extracts of *M. albicans*, determine their chemical composition, and evaluate their cytotoxicity and anti-inflammatory activity in different cells using in vitro models.

## 2. Results and Discussion

Recent works have investigated the composition of extracts obtained from the leaves of *M. albicans* [12]; however, little is known about its overall composition. In a previous investigation, phenolic compounds of *M. albicans* leaves were well characterized, identifying various flavonoids, including quercetin and several glycosylated derivatives, such as rutin, along with ellagic acid derivatives [8]. Phenolic compounds are known for their potential benefits to human health, mainly related to their antioxidant capabilities [13]. On the other hand, some phenolics are known to be cytotoxic; therefore, plant extracts rich in phenolics should be investigated for safety [14].

To perform a comprehensive phytochemical evaluation, the leaves of *M. albicans* were extracted in two different solvents: water, and 70% ethanol (Figure 1).

Although the two extraction media have different properties, the composition of the obtained extracts was quite similar when assessed by LC–MS, differing mainly in the individual components’ abundances (Figure 2). Moreover, to investigate compounds that can exert beneficial or hazardous effects, the crude extracts were fractionated in a liquid–liquid partition, yielding four fractions from each extract (aqueous and ethanolic): chloroform (Aq-Ch/Et-Ch), ethyl acetate (Aq-EtAc/Et-EtAc), butanol (Aq-But/Et-But), and the residual fractions (Aq-Res/Et-Res).

The yield of each fraction is represented in Table 1, and their chromatograms (Figure 2) show that the fractioning process was effective in separating the compounds based on their polarities. The total yields of the aqueous and ethanolic extracts were quite similar; however, when liquid–liquid fractionation was performed, the higher yield was obtained for solvent fractions originating from the ethanolic extract.

The contents of phenolic compounds of the aqueous and ethanolic extracts were determined using Folin–Ciocalteu reagent, and the observed values were 384 and 420 mg of gallic acid equivalents/g of dry extract, respectively. In a study carried out by Pieroni et al. (2011) [15], it was found that methanolic extract of *M. albicans* leaves presented 70.04 ± 0.12 mg of gallic acid/g of dry extract, while in a study by Lima et al. (2020) [12], the authors found 551.3 ± 3.72 mg of gallic acid/g of dry extract in their ethanolic extract. This difference in the total phenolic contents between the studies can be justified for two reasons: the solvent used for the extraction process, and the method of extraction.

### 2.1. Chemical Evaluation of M. albicans Solvent Fractions

The main constituents of both crude extracts obtained in this study were flavonoids, including their O-glycosides; nevertheless, compounds that have not been previously described were found in this study. The first mass spectrometry stage (MS^1^) analyses were performed in the positive and negative ionization modes, and the negative polarity gave poor results; therefore, only the analysis of protonated adducts [M+H]+ was considered. The second mass spectrometry stage (MS^2^) produced the fragmentation spectra from components of the crude extracts (Table 2), indicating the presence of quinic acid (peak 1) at *m*/*z* 193.0718, and of gallic acid at *m*/*z* 171.0297 (peak 2). Peak 3, at *m*/*z* 202.1462, was not identified. Peak 4 was found at *m*/*z* 355.1039, being consistent with chlorogenic acid (or its isomer), while peak 5, at *m*/*z* 291.0874, was consistent with catechin, which was further confirmed with an authentic standard. Peak 6, at *m*/*z* 387.2042, gave the fragments at *m*/*z* 225.1505, 207.1401, and 95.0866. Although the neutral loss (NL) of 162.054 strongly suggests the loss of a hexosyl residue (e.g., glucose), the aglycone with the fragment at *m*/*z* 225.1505 was not confirmed.

Different flavonol glycosides were found in this study, mainly containing quercetin as a glycone moiety originating from the protonated fragment at *m*/*z* 303.05, and myricetin was also found, as confirmed by the fragment at *m*/*z* 319.04. The most common monosaccharides attached to the flavonols were hexoses (e.g., glucose and/or galactose), deoxyhexose (rhamnose), and pentoses (e.g., arabinose and/or xylose) [13]. The flavonol glycosides observed in our extracts had similar characteristics, with peak 7, at *m*/*z* 617.1177, identified as quercetin-galloyl-hexoside. Fragments confirming this structure were found at *m*/*z* 315.0743, consistent with a galloyl-hexose residue, after the loss of the quercetin moiety that appeared at *m*/*z* 303.0529, whereas the fragments at *m*/*z* 171.0305 and 153.0200 were consistent with the gallic acid moiety [8].

Peaks 8 and 9 were isomers, appearing at *m*/*z* 611.16, with a similar fragmentation profile at *m*/*z* 465.10, consistent with the loss of a rhamnosyl residue, with a neutral loss (NL) of 146.05 atomic mass units (a.m.u.). The fragment at *m*/*z* 303.05, consistent with quercetin, was produced after an NL of 162.05 a.m.u. (from *m*/*z* 465.10 → 303.05), confirming the loss of a hexosyl residue (e.g., glucose or galactose). Peaks 8 and 9 had similar structures to that of rutin; however, this information was discarded when compared with the authentic standard. Therefore, peaks 8 and 9 were consistent with rutin isomers, as described in other plant extracts [13].

Peak 10 gave a protonated ion at *m*/*z* 465.1058, with a main fragment at *m*/*z* 319.0489, consistent with myricetin aglycone. The NL was 146.056, consistent with the loss of a rhamnosyl residue. Quintanas-Junior et al., in 2020 [8], also detected myricetin glycoside in extracts of *M. albicans*. Peak 11 appeared at m/z 539.2146, giving rise to fragments at *m*/*z* 315.0735, 207.1397, 171.0300, 153.0194, and 127.0389. This compound was not identified; however, the fragment at *m*/*z* 315.0735 was consistent with a galloyl-hexoside residue, while those at *m*/*z* 171.0300 and 153.0194 were from the gallic acid moiety, with this compound being consistent with a hydrolysable tannin. Peak 12, at *m*/*z* 611.1634, seemed to be an isomer of peaks 8 and 9; however, a fragment at *m*/*z* 449.109 was consistent with a rhamnosyl-quercetin, since the aglycone appeared at *m*/*z* 303.0533. Therefore, the glycan moiety observed on peak 12 had a different sequence from those observed in 8 and 9, consistent with a quercetin-hexosyl-rhamnoside [13].

Peak 13 appeared at *m*/*z* 435.0521, with a main fragment at *m*/*z* 303.0522 from quercetin. The NL was 132 a.m.u., consistent with the loss of a pentosyl residue (typically from arabinose or xylose) [13]. Peak 14, at m/z 581.1516, showed fragments at *m*/*z* 449.1080 from the loss of a pentosyl residue, and a fragment at *m*/*z* 303.0521 from quercetin after the loss of a rhamnosyl unit. A similar structure was found in *Maytenus ilicifolia* [13]. Peak 15 appeared at *m*/*z* 465.1044, with a main fragment at *m*/*z* 319.0472, consistent with a myricetin-rhamnoside [8]. Peak 16 was identified as a quercetin-rhamnoside, at *m*/*z* 449.1123, with a main fragment at *m*/*z* 303.0527; meanwhile, peak 17 was identified as free quercetin, at *m*/*z* 303.0511, as confirmed with an authentic standard.

Peak 18, at *m*/*z* 629.1528, was not identified, but the fragments suggested a phenolic compound, whereas peak 19, at *m*/*z* 615.3907, presented a fragment at *m*/*z* 453.3377, with an NL of 162.053, consistent with the loss of a hexosyl residue. Other fragments observed were consistent with a triterpenoid nucleus (Table 2), suggesting a monoglycosyl-saponin. Peak 20 appeared at *m*/*z* 345.0629; this compound had a similar UV absorbance to that of flavonols (255 and 360 nm), and a fragment observed at *m*/*z* 330.0387 was consistent with a radical cleavage, with the loss of CH3• (NL 15.024 a.m.u.), indicating a methoxylated flavonol. Assisted by the LIPID MAPS^®^ Database (https://www.lipidmaps.org/databases/lmsd/LMPK12113260?&LMID=LMPK12113260, accessed on 2 September 2022), the compound that matched with the MS data observed in this study was a trihydroxy-methoxy-methylenedioxyflavone, such as wharangin, which has been found in other vegetables, as described by Azizah et al. (2020) [16]. Peak 21, at *m*/*z* 329.0310, did not produce any fragments, preventing its identification. The LC–MS data and the compound identification are summarized in Table 2.

The compound distribution among the fractions is presented in Table 3. It was observed that aqueous and ethanolic extracts were composed of the same compounds and same amounts of each component. The fractionation with solvents generated four fractions (i.e., chloroform, ethyl acetate, butanol, and residual fractions) for each extract, which presented close similarity. No significant differences were observed when the water-derived fraction was compared to the correspondent ethanol-derived fraction. Therefore, Table 3 presents the relative distribution of flavonoids and other chemicals for both extracts.

**Table 2 molecules-27-05954-t002:** Characterization of the compounds identified by HPLC–MS.

Peak	Rt	MS ^1^	MS ^2^	Tentative Identification	Ref
**1**	2.5	193.0718	172.097, 147.066, 139.039, 129.054	Quinic acid	[13]
**2**	4.5	171.0297	125.0240, 107.0128, 81.0334	Gallic acid	[13]
**3**	7.2	202.1462	146.0830, 100.0771	N.I.	
**4**	7.4	355.1039	289.0727, 259.0602, 235.0617, 205.0507, 165.0557, 99.0632, 73.0467	Chlorogenic acid	[13]
**5**	8.4	291.0874	207.0666, 165.0553, 147.0450, 139.0400, 123.0449	Catechin	[13]
**6**	8.9	387.2042	225.1505, 207.1401, 95.0866	Glycoside	
**7**	9.6	617.1177	315.0743, 303.0529, 171.0305, 153.0200	Quercetin-galloyl-hexoside	[13]
**8**	9.7	611.1636	465.1056, 303.0527, 147.0666	Quercetin-rhamnosyl-hexoside	[13]
**9**	9.8	611.1604	465.1056, 303.0527, 147.066	Quercetin-rhamnosyl-hexoside	[13]
**10**	10.0	465.1058	319.0489, 245.0444, 217.0511, 153.0200,	Myricetin-rhamnoside	
**11**	10.1	539.2146	315.0735, 207.1397, 171.0300, 153.0194, 127.0389	Hydrolysable tannin	[17]
**12**	10.3	611.1634	449.109, 345.0618, 303.0533, 147.0667	Quercetin-hexosyl-rhamnoside	[13]
**13**	10.7	435.0421	303.0522, 225.0771, 147.0663, 97.0288	Quercetin-pentoside	[13]
**14**	10.9	581.1516	449.1080, 303.0521, 225.0770, 147.0662	Quercetin-pentosyl-rhamnoside	[13]
**15**	11.0	465.1044	319.0472, 129.0552	Myricetin-rhamnoside	[18]
**16**	11.1	449.1123	303.0527, 229.0510, 153.0187	Quercetin rhamnoside	[13]
**17**	14.4	303.0511	229.0517, 153.0186, 68.9959	Quercetin	
**18**	15.0	629.1528	423.0567, 277.0364	N.I.	
**19**	15.4	615.3907	537.2719, 453.3377, 407.3329	Triterpenoid hexoside	
**20**	17.9	345.0629	330.0387, 287.0217, 199.0388, 157.0864, 99.0443	Trihydroxy-methoxy-methylenedioxy-flavone	[16]
**21**	18.5	329.0310		N.I.	

^1^ 1st MS stage. ^2^ 2nd MS stage (fragmentation). N.I. = not identified in the positive ionization mass spectrometry.

**Table 3 molecules-27-05954-t003:** Compounds’ distribution among the different fractions.

Peak	Compound	Res ^a^	But ^a^	EtAc ^a^	Ch ^a^
**1**	Quinic acid	+++	++	-	-
**2**	Gallic acid	-	++	+++	-
**3**	N.I.	++	-	++	-
**4**	Chlorogenic acid	++	+	-	-
**5**	Catechin	+	+	-	-
**6**	Glycoside	+	+	+	-
**7**	Quercetin-galloyl-hexoside	-	++	+	-
**8**	Quercetin-rhamnosyl-hexoside	-	+	-	-
**9**	Quercetin-rhamnosyl-hexoside	-	+	-	-
**10**	Myricetin-rhamnoside	-	+	+	-
**11**	Hydrolysable tannin	-	+	+	-
**12**	Quercetin-hexosyl-rhamnoside	+	+++	++	-
**13**	Quercetin-pentoside	+	++	++	-
**14**	Quercetin-pentosyl-rhamnoside	-	+++	+	-
**15**	Myricetin-rhamnoside	-	++	+++	-
**16**	Quercetin rhamnoside	-	+	+++	-
**17**	Quercetin	-	+	+	+
**18**	N.I.	-	-	++	+
**19**	Triterpenoid hexoside	-	-	-	+
**20**	Trihydroxy-methoxy-methylenedioxy-flavone	-	+	-	++
**21**	N.I.	-	+	-	+++

^a^ Fractions were derived from aqueous and ethanolic extracts after fractionation using different solvents: chloroform (Ch), ethyl acetate (EtAc), *n*-butanol (But), and residual fraction (Res). The relative abundance of each compound is represented as follows: absent (-); low concentration (+); medium concentration (++); high concentration (+++).

### 2.2. Cytotoxicity Evaluation of the Extracts and Solvent Fractions

The toxicity of *M. albicans* was poorly verified up to this moment, and considering that this plant is recommended to be consumed daily in folk medicine, it is extremely important to verify the effects of *M. albicans* extracts and some of their main compounds via cytotoxic assays. For this purpose, two cell lines were used—Vero (normal monkey kidney epithelial cells) and Hep-G2 (human hepatocarcinoma cells)—as models to assess the cytotoxicity of the aqueous and ethanolic extracts, and all derived fractions, using the PrestoBlue^®^ assay. The results showed that crude extracts and butanol fractions presented low toxicity when the concentration was up to 100 µg/mL, inhibiting cell growth by less than 20% in both cell lines during the evaluated periods (24 h and 48 h) (Figure 3 and Figure 4). However, at higher concentrations (750 and 1000 µg/mL), some extracts—such as ethanol and its *n*-butanol fraction (Et-But)—were toxic to cells. The Vero cell line was more sensitive to treatments compared to Hep-G2 (Figure 3 and Figure 4), which can be explained by Hep-G2’s malignant cell characteristics. Furthermore, the chloroform (Aq-Ch and Et-Ch) and ethyl acetate (Aq-EtAc and Et-EtAc) fractions were highly toxic, especially to the Vero cell line (Appendix A). The toxicity of the non-polar fractions (Ch and EtAc) can be explained by the absence (Ch) or lower concentration (EtAc) of flavonoids and the presence of terpenoids, flavones, and non-identified compounds (Table 3).

It is important to note that the aqueous extract (Aq) and the Aq-But fraction showed almost no toxicity to either cell line, even after 48 h of treatment (Figure 3 and Figure 4). The Aq-But fraction presented many flavonoids, such as quercetin, gallic acid, and quercetin isomers, as can be seen in the HPLC chromatogram and Table 3. Such compounds can act as chemoprotective adjuvants, and their properties include the well-known antioxidant activity and free radical scavenging [13], which could explain the low toxicity of the fractions. Furthermore, there are few studies on *M. albicans* extracts and their properties, and none of them have reported any evaluation on cell lines; therefore this is the first report about the cytotoxicity of *M. albicans* extracts to the Vero and Hep-G2 cell lines, demonstrating that the fractionation of aqueous and ethanolic compounds may modify or intensify their cytotoxic (chloroform- and ethyl-acetate-derived fractions) or non-toxic (crude and butanol-derived fractions) properties through the separation and concentration of flavonoids from other phytochemicals present in the plant.

### 2.3. Redox Potential of M. Albicans Extracts

The phenolic compounds of plants are usually related to their antioxidant activity, because they possess redox properties. Considering the low toxicity of the aqueous, ethanolic, and butanol fractions on the tested cell lines, the potential antioxidant activity of such samples was evaluated by DPPH and TBARS assays. Reactive oxygen and nitrogen species (ROS and RNS) are continuously produced by cells as part of their metabolism. Imbalance between their synthesis and clearance is often associated with a lack of free radical scavengers, referred to as antioxidant molecules. The excess of reactive species inflicts several types of damage to cells and tissues, being associated with the pathogenesis of several diseases [19]. As expected, the four tested fractions (Aq, Et, Aq-But, and Et-But) exhibited marked DPPH scavenging activity, with IC_50_ values of 54.78 and 48.55 μg/mL for the aqueous and ethanolic extracts, respectively (Figure 5). Importantly, such concentrations were not toxic to the Vero and Hep-G2 cell lines, as previously described. Interestingly, the antioxidant activity observed for the butanol fractions was lower than that observed for the crude extracts. In the study performed by Lima et al. (2020) [12], an ethanolic extract of *M. albicans* produced by turbo-extraction presented DPPH radical scavenging activity that was increased in a concentration-dependent manner, and showed an IC_50_ of 33.86 μg/mL. Previously, Pieroni et al. (2011) [15] investigated the antioxidant activity of a methanol extract and the *n*-butanol fraction of *M. albicans* leaves. The *n*-butanol fraction was more bioactive than the methanol extract, showing IC_50_ values of 7.72 ± 0.035 and 49.45 ± 0.005 μg/mL, respectively. The DPPH scavenging activity of the crude extracts in the present study was similar to the activity of the methanol extract produced by Pieroni et al. (2011) [15]. However, the only compound present in all of the extracts was quercetin. Therefore, no conclusions could be reached about which compounds promote a greater antioxidant effect.

Since lipid peroxidation is related to many pathological processes, and plays an important role in biological systems, some of the *M. albicans* extracts were evaluated using the TBARS assay to determine whether they could influence this free-radical-mediated chain of reactions. The results are shown in Figure 6, and demonstrate that the aqueous extract was able to inhibit lipid peroxidation by 1.5× at 25 µg/mL, compared to the control with no antioxidant agents. The Aq-But fraction, in turn, showed similar inhibition (1.8×) at 50 µg/mL, while the ethanolic extract and its derived butanol fraction (Et-But) presented inhibition of lipid peroxidation at 100 µg/mL. The four evaluated fractions presented similar compositions; however, they varied in the quantity of each flavonoid, which could have caused the differences observed in their antioxidant activity. In the study by Lima et al. (2020) [12], the results of inhibition of lipid peroxidation in rat brain homogenates with *M. albicans* ethanol extract, produced via turbo-extraction, were able to inhibit lipid peroxidation at all evaluated concentrations (3–1000 µg/mL). Despite being a different model from the one tested in this study, this also showed a positive result, indicating that *M. albicans* extracts can act against lipid peroxidation.

### 2.4. Evaluation of M. albicans Extracts on THP-1 Macrophages

The effects of *M. albicans* extracts and fractions were evaluated by PrestoBlue^®^ assay on the THP-1 cell line to verify any possible harmful effect on these macrophages. The results are shown in Appendix A, and indicate that the tested extracts demonstrated a slight cytotoxic effect on THP-1 macrophages at the highest concentrations (750 and 1000 µg/mL), without drastically changing the cells’ viability.

Along with the inflammation process, reactive oxygen species (ROS) are involved in both cell proliferation and survival, as well in the induction of oxidative stress and apoptosis in tumoral cells [20].

The cartilage from patients with OA produces micromolar concentrations of NO, whereas normal cartilage does not produce NO, unless it is stimulated with TNF-α or IL-1β [19]. NO contributes to the pathophysiology of OA through different activities leading to the degradation of cartilage [21], and its removal is extremely important to control inflammatory process.

Using the DCFH-DA assay, an evaluation of the extracts’ ability to reduce the formation of ROS was performed, and the results are presented in Figure 7. The results showed that there was a marked reduction in ROS production after 12 h of treatment with all evaluated fractions (Aq, Et, and Aq-But), even when THP-1 macrophages were incubated with LPS plus extracts. Treatment with 50 µg/mL of Et and Aq-But decreased the LPS-induced ROS production by more than 4x, while the aqueous extract (Aq) required a concentration of 250 µg/mL to present the same effect. These data show that *M. albicans* extracts induce potent antioxidant activities in macrophages, which is consistent with the findings of Lima et al. (2020) [12], who evaluated the production of free radicals using a different methodology, and observed that the extract was able to reduce the production of nitrite in a range of 1–100 μg/mL (*p* < 0.001). The presence of *O*-glycoside flavonoids in the extracts—such as quercetin-galloyl-hexoside, quercetin-hexosyl-rhamnoside, quercetin-pentoside, and quercetin-pentosyl-rhamnoside—in higher proportions, as can be observed in Table 3, confirms the antioxidant properties, since it is well known that flavonoids present such properties [10].

To evaluate the anti-inflammatory activity of *M. albicans* extracts, we evaluated the levels of the pro-inflammatory cytokines TNF-α and IL-1β and the anti-inflammatory cytokine IL-10 in the supernatants of macrophages treated with the extracts obtained from this plant (Aq, Et, and the Aq-But fraction).

The results showed that the treatment with the Aq-But fraction was able to significantly reduce the levels of TNF-α induced by LPS after 48 h of treatment at 250 µg/mL (Figure 8A–C), while the crude extracts had a slight effect after 72 h (data not shown). When the effect on IL-1β production was evaluated (Figure 8D–F), the Aq-But fraction again proved to be the most effective in reducing the production of this pro-inflammatory cytokine. After 24 h of treatment with Aq-But at 50 µg/mL, the LPS-induced levels of this interleukin significantly decreased, while treatment with the crude extracts required a higher concentration (250 µg/mL) to achieve the same effect (Figure 8D–F). These results corroborate the study of Quintans-Júnior et al. (2020) [8], who observed a reduction in the main pro-inflammatory cytokines and chemokines related to arthritis (i.e., TNF-α, IL-1β, IL-6, and IL-8) using a murine model of CFA-induced rheumatoid arthritis (Freund’s complete adjuvant). Considering that TNF-α and IL-1β are abundant in patients with OA, and that the best therapies for this disease are based on reducing or modulating these cytokines to decrease some of the symptoms of the disease [22], it could be inferred that *M. albicans* extracts could have a beneficial effect on OA patients.

On the other hand, the IL-10 cytokine, which tends to increase in anti-inflammatory environments [23], did not show any differences after treatment with *M. albicans* extracts during the evaluated incubation periods (Figure 9).

The results observed in the anti-inflammatory experiments demonstrate that the Aq-But fraction probably maintained the compounds that present no toxicity and act as anti-inflammatory agents, unlike the chloroform- and ethyl-acetate-derived fractions, which presented triterpenoids and some unidentified compounds in high proportions, which could cause harm to the cells (Table 3). More studies are required to determine the mechanisms of action and to identify the main compounds that could cause the anti-inflammatory activity, as well as to identify the main toxic compounds that should be avoided by people who use this plant as primary healthcare phytotherapy.

## 3. Material and Methods

### 3.1. Chemicals

The solvents ethyl alcohol, methyl alcohol, chloroform, ethyl acetate, and *n*-butanol were purchased from Merck (Sao Paulo, Brazil). The chromatographic solvents acetonitrile and formic acid were HPLC grade, and were obtained from Sigma-Aldrich^®^ (Sao Paulo, Brazil). Type 1 ultrapure water was used for all experiments (Milli-Q, Merck Millipore, Burlington, MA, USA). The authentic standard of catechin, RPMI 1640 and Dulbecco’s Modified Eagle’s Medium/Ham’s Nutrient Mixture F12 (DMEM F-12) culture media, and phorbol 12-myristate 13-acetate (PMA) were purchased from Sigma-Aldrich^®^ (Sao Paulo, Brazil), while fetal bovine serum and penicillin/streptomycin were purchased from Gibco^®^ (Waltham, MA, USA).

### 3.2. Plant Material

*M. albicans* leaves were obtained from the herbarium of *Universidade Estadual de Ponta Grossa*, Ponta Grossa, Paraná, Brazil, and a voucher specimen was deposited at the same herbarium under the number UPCB 67694 (lat: −24.133333 long: −49.633333 WGS84). The studied species was registered on the *Sistema Nacional de Gestão do Patrimônio Genético e do Conhecimento Tradicional Associado* (SISGEN), number A2B17CB.

### 3.3. Miconia albicans Extracts

Two different extracts were prepared: the aqueous extract was obtained using 150 g of dried and ground leaves of *M. albicans* in 300 mL (3×) of ultrapure water, at 100 °C for 1 h, under reflux, while the ethanol extract was obtained after infusing 150 g of sample with 70% ethanol (300 mL, 3×) at 60 °C, for 1 h, under reflux (Figure 1). Both extracts were filtered and concentrated under reduced pressure, at 60 °C, using a rotary evaporator, until reaching 250 mL. An aliquot (50 mL) of each was lyophilized to calculate yield, and the remaining volume was used for fractionation.

### 3.4. Fractionation of Extracts

The crude extracts (aqueous (Aq) and ethanol (Et)) were fractioned using the liquid–liquid partition method (Figure 1). Prior to this procedure, the aqueous extract underwent a precipitation treatment with ethanol (3 vol.%) to remove the components with a higher molecular weight, such as polysaccharides. After precipitation, the insoluble fraction was separated by centrifugation (5000 rpm), and the soluble content was concentrated to a small volume under reduced pressure. The ethanol extract followed the liquid–liquid partition directly.

The fractionation was carried out individually in both extracts (Aq (200 mL) and Et (200 mL)), which were mixed sequentially with organic solvents of different polarities [24]. The first solvent added was chloroform (200 mL), and the mixture was vigorously stirred and held until the phases were separated. The chloroform layer was collected, and the procedure was repeated thrice, yielding the Aq-Ch and Et-Ch fractions. To the remaining aqueous layer, 200 mL of ethyl acetate was added and fractionated similarly (3×) to yield the Aq-EtAc and Et-EtAc fractions. The aqueous layer was then fractionated with *n*-butanol (200 mL, × 3), yielding the fractions Aq-But and Et-But, along with the residual fractions Aq-Res and Et-Res.

### 3.5. Determination of Phenolic Compounds

The phenolic compounds present in the aqueous and ethanolic extracts were evaluated based on the Folin–Ciocalteau colorimetric method [25], with some modifications. For the assay, 5 µL of sample, plus 125 µL of 10% Folin–Ciocalteau reagent and 100 µL of 7.5% sodium carbonate (Na_2_CO_3_), was added in microtubes. The standard curve was prepared in distilled water using gallic acid at 0.09, 0.18, 0.375, and 0.75 µg/mL. The samples were solubilized in distilled water at 1 mg/mL. After incubation at 45 °C for 15 min, the absorbance was measured at 765 nm with a spectrophotometer (BioTek^®^ Epoch, Wiinooski, VT, USA). This experiment was performed in triplicate. The total phenolic compounds were expressed as gallic acid equivalents (mg of gallic acid/g of sample).

### 3.6. Phytochemical Analysis

The phytochemical characterization was performed by high-performance liquid chromatography, using a Prominence LC20A (Shimadzu, Kyoto, Japan), equipped with a quaternary pump, column oven, autosampler, and photodiode array detector. High-resolution mass spectrometry (HRMS) was performed using a Maxis Q-ToF spectrometer (Bruker Daltonics, Billerica, MA, USA). The separation was carried out via reverse-phase chromatography, using a C18 column, with 250 × 4.6 mm and 3 µm particles size (Phenomenex Luna), held at 40 °C. The mobile phase consisted of type 1 ultrapure water (Milli-Q, Merck-Millipore, Burlington, MA, USA) and HPLC-grade acetonitrile (LiChrosolv-Merck), both acidified with 0.1% formic acid (Tedia, 97%). The chromatography was developed in a gradient of solvents, at a flow rate of 1 mL/min, starting with 5% acetonitrile, increasing to 80% in 25 min, to 100% in 29 min, and returning to 5% in 30 min, before being held in the initial conditions for 5 min for system equilibration. Samples were detected by PDA (200–800 nm) or HRMS (*m/z* 100–1500). The MS analyses were performed in the positive and negative ionization modes. For positive ionization, the energies were set at 500 V in the end plate offset and 4.5 kV in the capillary. The source temperature was held at 250 °C, and nitrogen was used for the sample desolvation, with dry gas at 8 L/min and nebulizer pressure at 2 bar. The fragmentation was performed by collision-induced dissociation (CID) in a data-dependent analysis (DDA) mode, using argon as the collision gas, with a voltage ramp of 10–60 eV. The samples were prepared in methanol–water, injected at a volume of 10 µL. The tentative compounds were identified based on their fragmentation profile, or assisted by authentic standards chromatographed under the same conditions. Crude aqueous and ethanol extracts were evaluated, along with their fractions obtained by liquid–liquid partition.

### 3.7. Scavenging Activity of M. albicans Extracts Measured by DPPH Assay

The scavenging activity of the samples (Aq, Et, Aq-But, and Et-But) was evaluated using the diphenyl-1-picrilhidrazil (DPPH) radical, according to the method described by Kamdem et al. (2013) [26]. The assay was evaluated in triplicate, and the extracts were prepared in distilled water at concentrations of 25, 50, 100, 250, and 500 µg/mL, using 96-well polystyrene plates. In each well, 10 µL of sample, water (negative control), or ascorbic acid (positive control) was added. Then, 30 µL of DPPH (1 mM) and 160 µL of ethanol were added. The blank of each sample was performed by adding the same volume of each reagent, without DPPH. The plate was incubated at room temperature for 30 min in the dark. Subsequently, the absorbance was read at 517 nm using a microplate reader (BioTek^®^ Epoch, Winooski, VT, USA).

### 3.8. Antioxidant Activity of M. albicans Extract Measured by TBARS Assay

The antioxidant activity of *M. albicans* extracts (Aq, Et, Aq-But, and Et-But) was determined by thiobarbituric-acid-reactive substances (TBARS) assay according to the method of Pinho et al. (2017) [27], with minor modifications. The comparative analysis of iron-induced lipid peroxidation was performed in microtubes, which were filled with 40 µL of phospholipid (10 mg/mL), 10 µL of Tris-HCl (50 mM), 10 µL of sample (25, 50, 100, 250, and 500 µg/mL) or 10 µL of water (negative control), and 10 µL of FeSO_4_ (0.0015 M). After incubation at 37 °C for 30 min, 100 µL of pH 3.4 acetic acid buffer (0.27 M hydrochloric acid and 1.33 M glacial acetic acid) was added to the tubes. Subsequently, 100 µL of 0.6% thiobarbituric acid (TBA) was added to the test tubes, and 100 µL of water in the negative control. After incubation at 100 °C for 1 h, 400 µL of butanol was added, and the tubes were vortexed for 30 s. The tubes were centrifuged at 6000 rpm for 10 min, and the supernatants were transferred to 96-well polystyrene plates to read the absorbance at 532 nm, using a microplate reader (BioTek^®^ Epoch).

In parallel, a standard malondialdehyde (MDA) curve was performed to quantify the production of TBARS. Four microtubes containing distilled water (70 µL, 60 µL, 50 µL, and 30 µL), MDA (0 µL, 10 µL, 20 µL, and 40 µL, respectively), TBA (100 µL), and acetic acid buffer pH 3.4 (100 µL) were used. After incubation at 100 °C for 1 h, 400 µL of butanol was added, and the tubes were vortexed for 30 s. The tubes were centrifuged at 6000 rpm for 10 min and the supernatants were transferred to 96-well polystyrene plates to read the absorbance at 532 nm, using a microplate reader (BioTek^®^ Epoch).

### 3.9. Cell Culture

The human monocytic cell line THP-1 was grown in RPMI 1640 culture medium supplemented with 10% bovine fetal serum and 1% penicillin/streptomycin (P/S) at 37 °C, in 5% CO_2_, in a humidified incubator. The medium was renewed twice per week. The Vero (animal kidney cells) and Hep-G2 (hepatocellular carcinoma cell line) cell lines were grown in DMEM F-12 culture medium, supplemented with 10% (*v*/*v*) fetal bovine serum and 1% penicillin/streptomycin, under a temperature of 37 °C and 5% CO_2_. The medium was renewed twice per week.

### 3.10. THP-1 Macrophage Differentiation

Macrophages were obtained from the treatment of THP-1 monocytes with 62.5 ng/mL phorbol 12-myristate 13-acetate (PMA) and maintained for 48 h at 37 °C under 5% CO_2_ in a humidified incubator. Subsequently, the medium containing PMA was removed, and the cells were washed with PBS. Fresh supplemented RPMI medium was added to the adhered and differentiated macrophages for further experiments.

### 3.11. Cytotoxicity of M. albicans Extracts on THP-1, Vero, and Hep-G2 Cells

To measure cell viability, the PrestoBlue^®^ reagent (resazurin) was reduced to resorufin by metabolically active cells, indicating mitochondrial activity of living cells. For THP-1, 4 × 10^4^ cells were plated per well in a 96-well plate (200 µL/well), and differentiated as described in Section 3.10. Vero and Hep-G2 cells were prepared in DMEM medium, and 3.5 × 10^4^ cells (200 µL/well) were also plated in a 96-well plate for a period of 24 h for adhesion. The three cell lines were treated with *M. albicans* fractions (20 µL), and vehicles and controls were added at the same volume, according to the solvent used to dilute the samples. After the incubation period (24 h and 48 h) at 37 °C, the entire medium was removed from the plates, and 90 µL of fresh medium plus 10 µL of PrestoBlue^®^ was added, followed by incubation for 1.5 h at 37 °C. The absorbance was read at 570 nm and 600 nm with a microplate reader (BioTek^®^ Epoch). The assays were carried out in three independent experiments, in quintuplicate. Cell viability was analyzed by subtracting the absorbances of the blanks, and expressed as a percentage of viable cells.

### 3.12. Evaluation of Oxidative Stress after Treatment with M. Albicans Extracts

The sensitive and rapid quantitation of reactive oxygen species in response to oxidative metabolism in cells can be determined using 2′,7′-dichlorofluorescin diacetate (DCFH-DA), which is a cell-permeable non-fluorescent probe. Upon oxidation, this compound is de-esterified intracellularly, and becomes highly fluorescent [28].

This assay was performed, as described by Drehmer et al. (2016) [29], with some modifications. Briefly, THP-1 cells (1 × 10^6^ cells/well) were seeded in 6-well culture plates and allowed to differentiate as described in Section 3.10. The cells were incubated with the extracts (Aq, Et, and Aq-But) (50 and 250 µg/mL), with LPS (500 ng/mL), or with “extracts + LPS”, for 12 h. Afterwards, 35 µM 2′,7′-dichlorofluorescin diacetate (DCFH-DA) (Sigma-Aldrich^®^) was added to each well, and the cells were incubated for 10 min at 37 °C, protected from light. The measurement of median fluorescence intensity (MFI) was performed using a FACS Canto II (Becton-Dickinson, BD^®^, Franklin Lakes, NJ, USA) flow cytometer.

### 3.13. Evaluation of Cytokines Released after Treatment with M. Albicans Extracts

The levels of the pro-inflammatory cytokines TNF-α (# 88-7346) and IL-1β (# 88-7261) and anti-inflammatory IL-10 (# 88-7106) were measured in the supernatants collected from THP-1 macrophages after incubation with *M. albicans* extracts, using the ELISA Kit from Invitrogen (Waltham, MA, USA), according to the manufacturer’s protocols. First, THP-1 cells were differentiated and treated as described in Section 3.10 and Section 3.12, for 24 h, 48 h, and/or 72 h. PBS was used as a negative control. The absorbance was read at 450 nm, and the tests were carried out in two independent experiments, in quadruplicate.

### 3.14. Statistical Analyses

The results obtained from the independent experiments are expressed as the mean ± standard error (SEM). The statistics were performed using one-way analysis of variance (ANOVA) followed by Bonferroni’s test; *p* ≤ 0.05 was considered statistically significant. The 50% inhibitory concentration (IC_50_) was calculated for each datum, when a dose–response was observed, using GraphPad Prism 9.0 (GraphPad Software^©^, San Diego, CA, USA). The other statistical analyses and graphs were performed using the same software.

## 4. Conclusions

This study showed the extraction and preparation of fractions from the leaves of *M. albicans*—a plant commonly used in Brazilian folk medicine for the treatment of osteoarthritis. The aqueous and ethanolic extracts had a similar composition, with the presence of many flavonoids, including quercetin and its isomers, gallic acid, and myricetin. The crude aqueous and ethanolic extracts, as well as their *n*-butanol fractions, showed antioxidant activity, and demonstrated potential anti-inflammatory activity by reducing TNF-α and IL-1β in THP-1 macrophages. Furthermore, when the extracts were tested on models of animal kidney cells (Vero) and hepatocellular carcinoma cells (Hep-G2), low or no toxicity was observed for the crude extracts and their *n*-butanol fractions up to 500 µg/mL. On the other hand, the chloroform and ethyl acetate fractions contained high concentrations of compounds that were highly toxic—especially to the Vero cell line. This study demonstrates for the first time how the toxicity of a plant can be removed by fractionation and separation of its compounds. Furthermore, the antioxidant and anti-inflammatory properties of *M. albicans* were indeed verified using our in vitro model. Considering its composition and the effects on important mechanisms of the pathophysiology of joint diseases, it can be concluded that aqueous and *n*-butanol extracts derived from *M. albicans* should be deeply investigated to determine their mechanisms of action and the possible applicability of this plant as a phytotherapeutic drug.

## Figures and Tables

**Figure 1 molecules-27-05954-f001:**
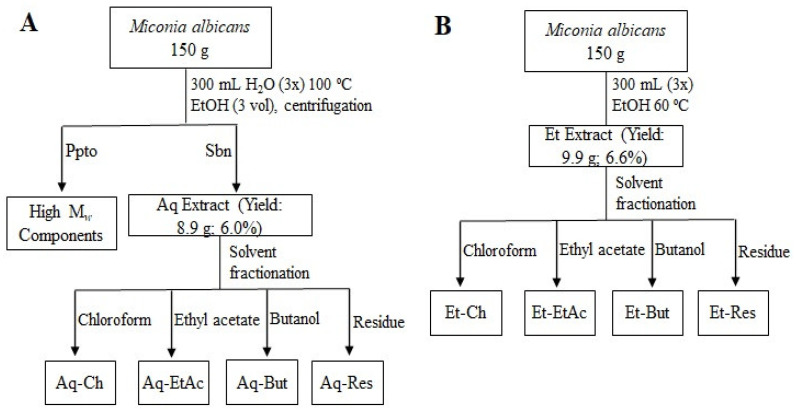
Extraction and fractionation scheme of *M. albicans* by aqueous (**A**) and ethanol (**B**) extraction.

**Figure 2 molecules-27-05954-f002:**
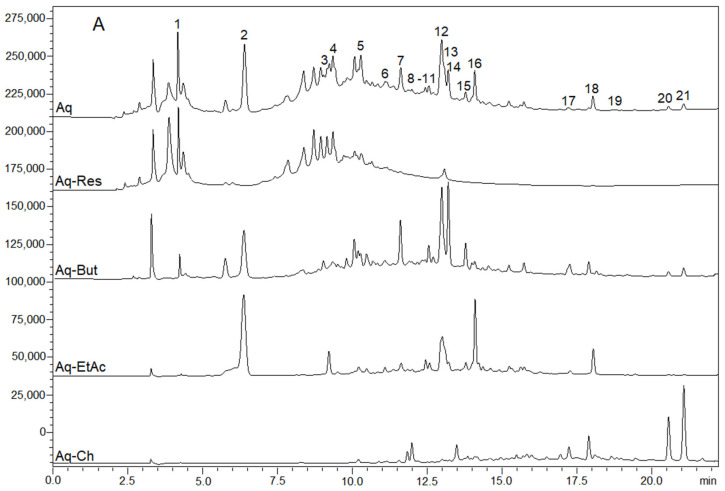
HPLC chromatographic profile of aqueous extract (Aq) and aqueous solvent fractions: chloroform (Aq-Ch), ethyl acetate (Aq-EtAc), *n*-butanol (Aq-But), and residual fraction (Aq-Res) (**A**); HPLC chromatographic profile of ethanolic extract (Et) and ethanolic solvent fractions: chloroform (Et-Ch), ethyl acetate (Et-EtAc), *n*-butanol (Et-But), and residual fraction (Et-Res) (**B**).

**Figure 3 molecules-27-05954-f003:**
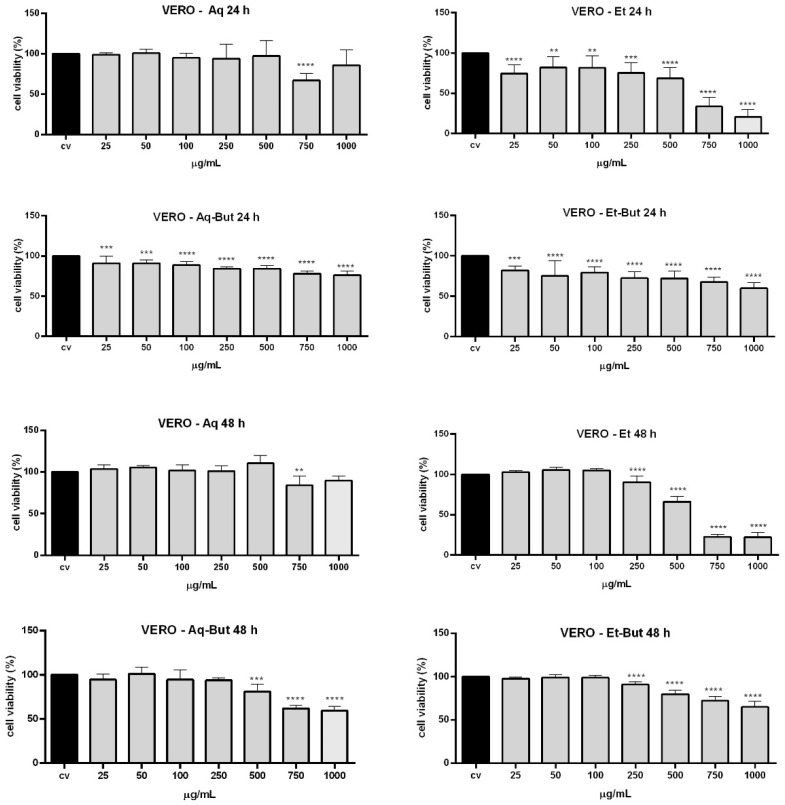
Cytotoxicity analysis of *M. albicans* extracts on Vero cells evaluated by PrestoBlue^®^. Cells were incubated with aqueous and ethanolic extracts, as well as their butanol fractions (25, 50, 100, 250, 500, 750, and 1000 µg/mL), for 24 h and 48 h. Statistical analyses were performed by one-way analysis of variance (ANOVA). followed by Bonferroni’s test, for selected pairs. Results represent the mean ± SD of two independent experiments (*n* = 2, in triplicate); ** *p* < 0.01, *** *p* < 0.001, **** *p* < 0.0001 versus vehicle control (V).

**Figure 4 molecules-27-05954-f004:**
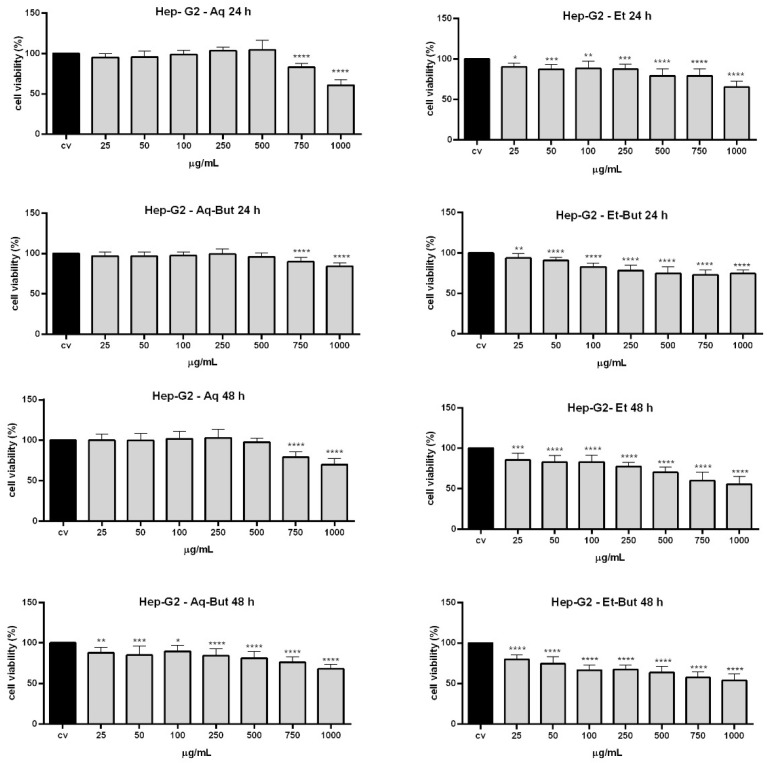
Cytotoxicity analysis of *M. albicans* extracts on Hep-G2 cells evaluated by PrestoBlue^®^. Cells were incubated with aqueous and ethanolic extracts, as well as their butanol fractions (25, 50, 100, 250, 500, 750, and 1000 µg/mL), for 24 h and 48 h. Statistical analyses were performed by one-way analysis of variance (ANOVA), followed by Bonferroni’s test, for selected pairs. Results represent the mean ± SD of two independent experiments (*n* = 2, in triplicate); * *p* < 0.05, ** *p* < 0.01, *** *p* < 0.001, **** *p* < 0.0001 versus vehicle control (V).

**Figure 5 molecules-27-05954-f005:**
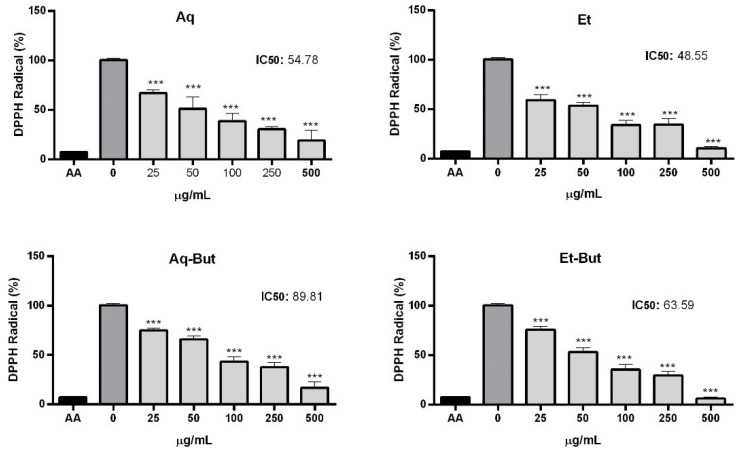
Analysis of the antioxidant effects of *M. albicans* extracts on DPPH radical scavenging. Aqueous (Aq) and ethanol (Et) extracts, as well as their *n*-butanol fractions (Aq-But, Et-But), were tested at 25–500 µg/mL. 0: negative control (vehicle). AA: positive control (ascorbic acid). Statistical analyses were performed by one-way analysis of variance (ANOVA), followed by Bonferroni’s test, for selected pairs. Results represent the mean ± SD of two independent experiments (*n* = 2, in triplicate); *** *p* < 0.001.

**Figure 6 molecules-27-05954-f006:**
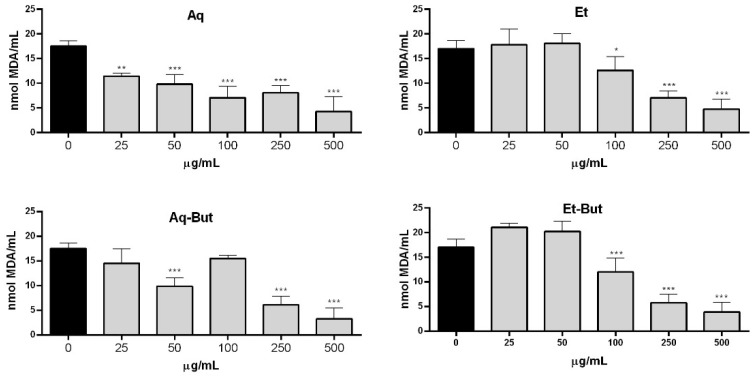
Analysis of the antioxidant effects of *M. albicans* extracts on lipid peroxidation induced by iron. Aqueous (Aq) and ethanol (Et) extracts, as well as their *n*-butanol fractions (Aq-But, Et-But), were tested at 25–500 µg/mL. Statistical analyses were performed by one-way analysis of variance (ANOVA), followed by Bonferroni’s test, for selected pairs. Results represent the mean ± SD of two independent experiments (*n* = 2, in quadruplicate); * *p* < 0.05, ** *p* <0.01, *** *p* < 0.001 versus control (0).

**Figure 7 molecules-27-05954-f007:**
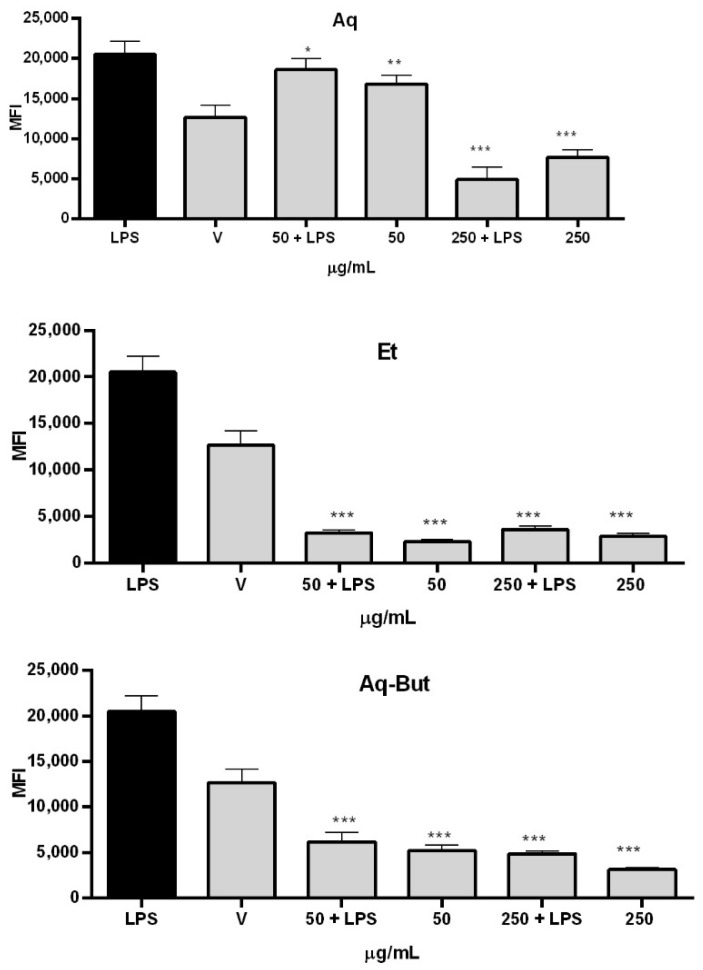
Reduction in ROS formation by THP-1 macrophages after treatment with *M. albicans* extracts, with or without LPS. Aqueous (Aq) and ethanol (Et) extracts, as well as the aqueous *n*-butanol fraction (Aq-But), were tested at 50 and 250 µg/mL, with or without LPS (500 ng/mL). Statistical analyses were performed by one-way analysis of variance (ANOVA), followed by Bonferroni’s test, for selected pairs. Results represent the mean ± SD of two independent experiments (*n* = 2, in triplicate); * *p* < 0.05, ** *p* < 0.01, *** *p* < 0.001 versus LPS.

**Figure 8 molecules-27-05954-f008:**
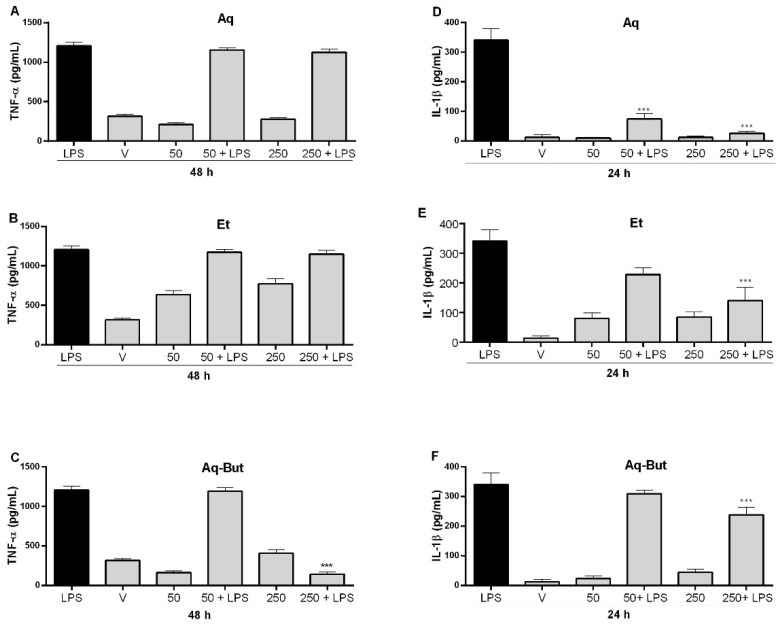
Effects of *M. albicans* treatment of THP-1 macrophages on the levels of TNF-α (**A**–**C**) and IL-1β (**D**–**F**). Statistical analyses were performed by one-way analysis of variance (ANOVA), followed by Bonferroni’s test, for selected pairs. Results represent the mean ± SD of two independent experiments (*n* = 3); *** *p* < 0.001 versus control (LPS).

**Figure 9 molecules-27-05954-f009:**
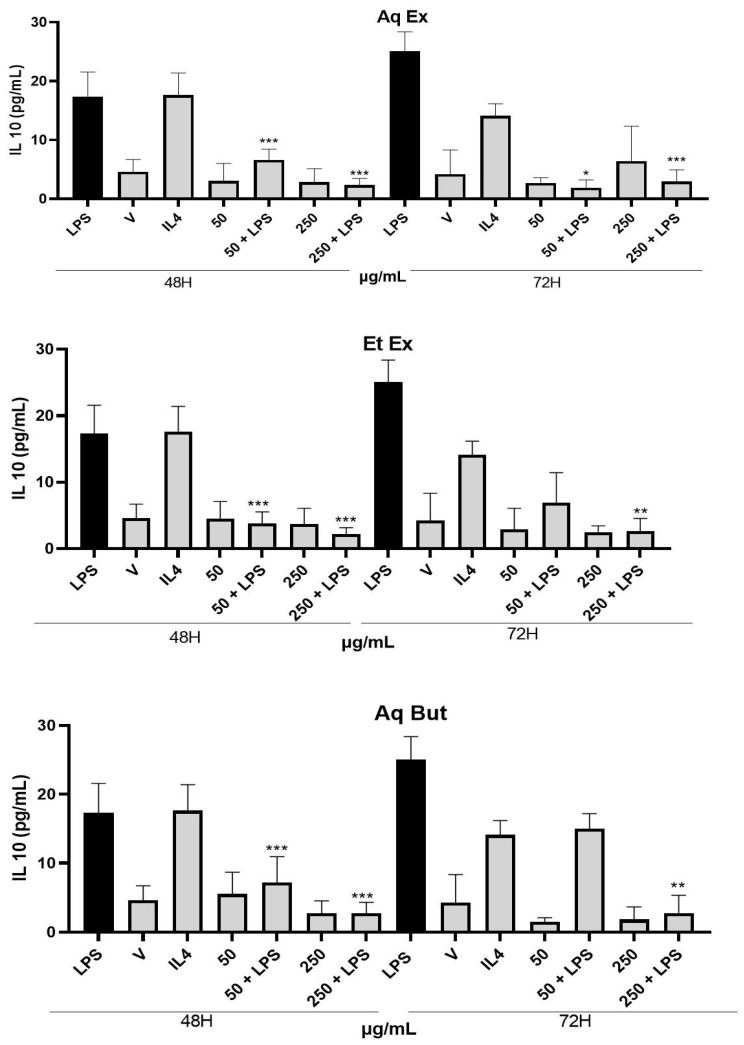
Effects of M. albicans treatment of THP-1 macrophages on the levels of IL-10. Statistical analyses were performed by one-way analysis of variance (ANOVA), followed by Bonferroni’s test, for selected pairs. Results represent the mean ± SD of two independent experiments (*n* = 2, in triplicate); * *p* < 0.05, ** *p* < 0.01, *** *p* < 0.001 versus control (LPS).

**Table 1 molecules-27-05954-t001:** Yields of aqueous and ethanolic extracts and their solvent fractions.

Fraction	Yield (%)
Aq	6.0 ^a^
Aq-Ch	0.9 ^b^
Aq-EtAc	4.6 ^b^
Aq-But	19.7 ^b^
Aq-Res	69.9 ^b^
Et	6.6 ^a^
Et-Ch	22.8 ^c^
Et-EtAc	10.1 ^c^
Et-But	27.8 ^c^
Et-Res	52.5 ^c^

^a^ Yield was calculated based on the dry weight of the material used for the initial extraction (150 g). ^b^ Yield was calculated based on the dry weight of the aqueous extract. ^c^ Yield was calculated based on the dry weight of the ethanolic extract.

## Data Availability

Not applicable.

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
