# Peer review of "Phytochemical Evaluation and Anti-Inflammatory Potential of Miconia albicans (Sw.) Triana Extracts"

_molecules, 2022, doi:10.3390/molecules27185954_

Round 1

Reviewer 1 Report

The paper shows a very detailed research regarding the extraction and characterization of Miconia albicans, the objective for this research is great since it is not only aimed to answer questions that lack answers, but it is also focused on solving a problem of plant availability to aid patients that suffer from OA. In general the research was thorough and complete; there are just some minor observations:

I do; however, have the following observations:

1) Here authors use Hep G2 (a human liver cancer cell line) and Vero (Cercopithecus aethiops kidney epithelial cells) these are two different species; is it really fair to compare the effects a place a statement that this could be expected in humans as well? why not use cell lines from the same species?

If possible, can Vero be replaced with a human fibroblast cell line?

Regarding the presentation of data, in my opinion its easier to view comparison of treatments when placed side by side and not on different charts

When viewing results from antioxidant activities, why not produce this analysis in cell lines and not just from a chemical point of view?

Author Response

We thank for the revision of this manuscript and for the suggestions that help us to improve the quality of this manuscript. All the requested observations were answered, and the responses are marked in blue. The corrections made to the manuscript are marked using the Track changes of WordMS.

Reviewer 1

Comments and Suggestions for Authors

The paper shows a very detailed research regarding the extraction and characterization of Miconia albicans, the objective for this research is great since it is not only aimed to answer questions that lack answers, but it is also focused on solving a problem of plant availability to aid patients that suffer from OA. In general the research was thorough and complete; there are just some minor observations:

I do; however, have the following observations:

1) Here authors use Hep G2 (a human liver cancer cell line) and Vero (Cercopithecus aethiops kidney epithelial cells) these are two different species; is it really fair to compare the effects a place a statement that this could be expected in humans as well? why not use cell lines from the same species?

Response: Indeed, the reviewer has an important observation about comparing the effect of the extracts on two cell lines from the same species. Our preliminary experiments were performed on human fibroblasts; however, the stock of such cells was contaminated and no longer could be used. As this study was performed during the pandemic situation (2020-2021), it was difficult to acquire new cell lines, therefore we decided to use Vero cell line that was available from our cell bank. We have chosen Vero cell line because of two reasons: (1) it is derived from the same Primate family (close to humans) and it is widely accepted as a model to evaluate toxicity; and (2) this cell line is more sensitive than human fibroblasts, therefore, if the extracts caused no toxicity to Vero cell line, it would probably cause no harm to other human cell line (Freire et al., 2009). Furthermore, we used THP-1 macrophages (a human monocytic cell line) and Hep-G2 (human hepatic cancer cell line) to determine the toxicity of the extracts. The results of cytotoxicity on THP-1 cells are presented on supplementary material (Fig S3).

Fernández Freire, P., Peropadre, A., Pérez Martín, J. M., Herrero, O., & Hazen, M. J. (2009). An integrated cellular model to evaluate cytotoxic effects in mammalian cell lines. Toxicology in Vitro, 23(8), 1553–1558. https://doi.org/10.1016/j.tiv.2009.06.017

If possible, can Vero be replaced with a human fibroblast cell line?

Response: At this moment we can not perform the experiments with human fibroblast cell line because we do not have such cell line in our cell bank. However, as explained above, we believe that the proximity of the two species (human and monkeys are included in the same Primate family) validate the results of this study. We have provided cytotoxicity of M. albicans extracts in THP-1 macrophages (a human monocytic cell line) and Hep-G2 (human hepatic cancer cell line) in addition to Vero evaluation.

Regarding the presentation of data, in my opinion its easier to view comparison of treatments when placed side by side and not on different charts

Response: We agree with the reviewer that, from some point of view, it is easier to compare the data if they are presented in the same graph. However, in this case, we would have to join 4 graphs at 1 and, therefore the graph would provide too much information and became difficult to visualize. Therefore, we decided to keep the graphs as they were presented in the original version.

When viewing results from antioxidant activities, why not produce this analysis in cell lines and not just from a chemical point of view?

Response: Actually, we have performed the DCFH-DA assay, which evaluates the capability of the extracts to reduce the formation of reactive oxygen species (ROS), on cells. This experiment was performed on the human monocytic cell line THP-1 and the results are presented in Figure 7. The legend of Figure 7 was not clear; therefore, the text was corrected and the information about the cell treatment was added.

Due to shortage of reagent delivery (during the 2020-2021 pandemic situation) we decided to measure ROS only on THP-1 cells because we did not have enough reagent to perform the experiments on the three cell lines.

Reviewer 2 Report

Authors present the extraction and investigation of fractions from the leaves of Miconia albicans, a known medicinal plant in Brazil in the folk medicine, containing many flavonoids and polyphenols. The chemically determined fractions were investigated for cytotoxic, antioxidant and anti-inflammatory activity.

This is an excellent work, however, need some other supplementary data.

1. It is not clear how it was decided which tables and figures should be included in the supplementary section or in the text.

2. Table S1 gives hardly any information without the proportion of the components

3. It would be important to know the exact polyphenol composition of the fractions, especially in the case of biological active ones.

4. IC50 values would be more appropriate, especially for cytotoxic activities of fractions showing valuable activity.

5. One more little thing: Table 2, peak 8: quercetin and no quercitin

Author Response

We thank for the revision of this manuscript and for the suggestions that help us to improve the quality of this manuscript. All the requested observations were answered, and the responses are marked in blue. The corrections made to the manuscript are marked using the Track changes of WordMS.

Reviewer 2

Comments and Suggestions for Authors

Authors present the extraction and investigation of fractions from the leaves of Miconia albicans, a known medicinal plant in Brazil in the folk medicine, containing many flavonoids and polyphenols. The chemically determined fractions were investigated for cytotoxic, antioxidant and anti-inflammatory activity.

This is an excellent work, however, need some other supplementary data.

  1. It is not clear how it was decided which tables and figures should be included in the supplementary section or in the text.

Response: Our decision about which figures should be in the main text and which ones should be in the supplementary material was based on: we did not see necessity to pollute the manuscript with information that is only provided to prove that internal control of the experiments was performed. Figures S1 and S2 presented data of cell viability (Vero and Hep-G2) of the two fractions which were toxic to the cells (chloroform and ethyl acetate derived fractions) and therefore were not used on the antioxidant and anti-inflammatory experiments. Figure S3 demonstrated that the chosen extracts (aqueous, ethanolic and butanol derived fractions) were not toxic to THP-1 macrophages, and therefore could be evaluated about their anti-inflammatory activity. Figure S4 was moved to the main text, named as Figure 9.

  1. Table S1 gives hardly any information without the proportion of the components

Response: Thanks for your observation. We agree that no important information could be extracted from the original Table S1. Therefore, we have changed this table, improving the symbols to indicate a relative proportion and distribution of each fraction component, as follows:

-     absent

+ – low concentration

++ – medium concentration

+++ – high concentration

Furthermore, this table was moved to the main text, as table 3.

It is important to note that these relative abundances were obtained by peak area integration, and not through a calibration curve, which should be prepared with the authentic standards for each component. Considering that there are no commercial standards available for each compound described, the exact quantification is unfeasible.

  1. It would be important to know the exact polyphenol composition of the fractions, especially in the case of biological active ones.

Response: The overall composition obtained by LC-MS is described in section 2.1 and Table 2. Figure 2 represents the original chromatograms of the extracts and fractions, containing the distribution of the phytochemical compounds in each fraction. Moreover, Table 3 brings now more information on the relative abundances of each fraction component. The study of plant extracts usually evaluates the biological effects of the whole extract, which is composed by a numerous of compounds that sometimes are not identified. In some cases, some natural compounds can be utilized as beginning points for drug discovery in chemical synthesis, serving as a foundation for the development of synthetic equivalents with a higher efficacy, potency, safety, and purity. However, the preliminary search for such compounds includes the overall screening of the biological activity of crude extracts. Our study is situated on this point of the research: preliminary screening and partial chemical characterization to understand if a plant used in folk medicine could be really employed for this purpose.

  1. IC50 values would be more appropriate, especially for cytotoxic activities of fractions showing valuable activity.

Response: Indeed, we agree that IC50 values are more appropriate in this case, however, the results obtained on the majority of the cytotoxic graphs (Aq, Et and Aq-But, Et-But) did not present values that could be attributed to ~0% viability or maximum toxicity. This effect is usually observed when crude plant extracts are tested because they are usually non-toxic or do not follow a dose-response curve. In this case, the IC50 calculation gives R2 ≠ 1 and therefore does not give reliable results. The IC50 of other graphs was calculated and a topic about this calculation was added to the “3.14 Statistical analyses” section.

  1. One more little thing: Table 2, peak 8: quercetin and no quercitin

Response: Ok, this was corrected.

Reviewer 3 Report

It's interesting to read Manzano et al study titled "Chemical evaluation of Miconia albicans (Sw.) Triana extracts and their potential as anti-inflammatory phytomedicine." However, with the considerable improvement I have listed below, please consider publishing this manuscript in the "Molecules" journal.

1.    Title: The investigation that was conducted in this manuscript is not accurately described by the title. Therefore, I advise the author to think about the following title, which might be appropriate. "Phytochemical evaluation and anti-inflammatory potential of Miconia albicans (Sw.) Triana extracts".

2.    Abstract: Future directions for this research are extended in the closing paragraph of the conclusion. For example, "In the future, it will be necessary to identify the phytochemicals that are responsible for anti-inflammatory effects for new drug discovery, development, and therapy against inflammation. In-vivo studies on M. albicans extracts is still required to confirm possible mechanisms of action.

3.    Introduction: Lines 41–46 should be cited to the most recent study on the subject (https://doi.org/10.3390/molecules27030734), in which the authors go into great detail regarding the pros and cons of the NSAIDs that are now in use. Add the most recent and relevant references for numbers line 51 as well (8,9). Add the name of the plant family on Line 53. Remove the word "some studies" from line 58 and replace it with "Previous/Earlier studies have shown."

4.    Figure 1 should show the main extracts yield in %.

5.    Calculate the IC50 value and add it in section 2.2.

6.    Since that the figures 6 and 7 contained the (*) indication, the legends of those figures must state *p<0.05.

7.    Remove **p<0.01 from Figure 8's legend because it wasn't included in Figure 8.

8.    Change Miconia albicans to M. albicans in Section 3.2

9.    Line 364, change the word “donated by the” to “obtained from the”

10.  Line 365, at 60 °C using what? Is a rotating evaporator used? Please don't omit it.

11.  Section 3.6. Change the word analyses to analysis

12.  Line 446, Outdated reference. Cite the most recent reference available for that process.

13.  Section 2 heading to be changed as “Results and Discussion”. Furthermore, the discussion needs to be improved. The author should provide a critical justification for the comparison of the results with the literature.

14.  Conclusions: The authors do not really provide any conclusions; they only give an overview of their findings. The conclusions and insights should be used to strengthen this section. As a consequence, everyone will understand the research outcome. The importance of this research should be highlighted by the author.

Author Response

We thank for the revision of this manuscript and for the suggestions that help us to improve the quality of this manuscript. All the requested observations were answered, and the responses are marked in blue. The corrections made to the manuscript are marked using the Track changes of WordMS.

Reviewer 3

Comments and Suggestions for Authors

It's interesting to read Manzano et al study titled "Chemical evaluation of Miconia albicans (Sw.) Triana extracts and their potential as anti-inflammatory phytomedicine." However, with the considerable improvement I have listed below, please consider publishing this manuscript in the "Molecules" journal.

  1. Title: The investigation that was conducted in this manuscript is not accurately described by the title. Therefore, I advise the author to think about the following title, which might be appropriate. "Phytochemical evaluation and anti-inflammatory potential of Miconia albicans (Sw.) Triana extracts".

Response: Thank you for the suggestion, we agree and changed the title as proposed.

  1. Abstract: Future directions for this research are extended in the closing paragraph of the conclusion. For example, "In the future, it will be necessary to identify the phytochemicals that are responsible for anti-inflammatory effects for new drug discovery, development, and therapy against inflammation. In-vivo studies on M. albicans extracts is still required to confirm possible mechanisms of action.

Response: Thank you for the suggestion, we agree with this suggestion and an inclusion of a closing paragraph was added to the abstract.

  1. Introduction: Lines 41–46 should be cited to the most recent study on the subject (https://doi.org/10.3390/molecules27030734), in which the authors go into great detail regarding the pros and cons of the NSAIDs that are now in use. Add the most recent and relevant references for numbers line 51 as well (8,9). Add the name of the plant family on Line 53. Remove the word "some studies" from line 58 and replace it with "Previous/Earlier studies have shown."

Response: Thank you for the suggestion, we believe that the references cited about the use of NSAIDs are recent (they are from 2019), however, we have added the new reference suggested from 2022 (https://doi.org/10.3390/molecules27030734) that discuss about the topic of anti-inflammatory drugs from plants. Furthermore, we have changed the references (8,9) for recent ones published in 2017 and 2020 (https://doi.org/10.3390/medicines7020007 and https://doi.org/10.1016/j.pop.2017.02.001).

The name of M. albicans family was added (Melastomataceae) and the text “some studies” was replaced by “earlier studies”.

  1. Figure 1 should show the main extracts yield in %.

Response: The yield in % was added to the aqueous and ethanolic extracts. The other extracts are described in table 1.

  1. Calculate the IC50 value and add it in section 2.2.

Response: Indeed, we agree that IC50 values are more appropriate in this case, however, the results obtained on the majority of the cytotoxic graphs (Aq, Et and Aq-But, Et-But) did not present values that could be attributed to ~0% viability or maximum toxicity. This effect is usually observed when crude plant extracts are tested because they are usually non-toxic or do not follow a dose-response curve. In this case, the IC50 calculation gives R2 ≠ 1 and therefore does not give reliable results. The IC50 of other graphs was calculated and a topic about this calculation was added to the “3.14 Statistical analyses” section.

  1. Since that the figures 6 and 7 contained the (*) indication, the legends of those figures must state *p<0.05.

Response: Thank you for the suggestion, it was corrected on both figures.

  1. Remove **p<0.01 from Figure 8's legend because it wasn't included in Figure 8.

Response: Thank you for the suggestion, it was corrected.

  1. Change Miconia albicans to M. albicans in Section 3.2

Response: Thank you for the suggestion, it was corrected.

  1. Line 364, change the word “donated by the” to “obtained from the”

Response: Thank you for the suggestion, it was corrected.

  1. Line 365, at 60 °C using what? Is a rotating evaporator used? Please don't omit it.

Response: Yes, it was used a rotating evaporator. This information was added.

  1. Section 3.6. Change the word analyses to analysis

Response: It was changed.

  1. Line 446, Outdated reference. Cite the most recent reference available for that process.

Response: An updated reference was added to replace the old one.

  1. Section 2 heading to be changed as “Results and Discussion”. Furthermore, the discussion needs to be improved. The author should provide a critical justification for the comparison of the results with the literature.

Response: Thank you for the suggestion, the discussion was improved, and we believe it is complete.

  1. Conclusions: The authors do not really provide any conclusions; they only give an overview of their findings. The conclusions and insights should be used to strengthen this section. As a consequence, everyone will understand the research outcome. The importance of this research should be highlighted by the author.

Response: Thanks, the conclusion was modified, and a closure idea was added to the text.

Round 2

Reviewer 3 Report

The authors addressed all of the comments and revised the content of the manuscript accordingly. In light of this, I recommend that it be published in Molecules Journal in its present form.